# The Reticulon-4 3-bp Deletion/Insertion Polymorphism Is Associated with Structural mRNA Changes and the Risk of Breast Cancer: A Population-Based Case–Control Study with Bioinformatics Analysis

**DOI:** 10.3390/life13071549

**Published:** 2023-07-12

**Authors:** Pouria Pourzand, Farhad Tabasi, Fariba Fayazbakhsh, Shamim Sarhadi, Gholamreza Bahari, Mohsen Mohammadi, Sahar Jomepour, Mohammad Nafeli, Fatemeh Mosayebi, Mehrdad Heravi, Mohsen Taheri, Mohammad Hashemi, Saeid Ghavami

**Affiliations:** 1Department of Clinical Biochemistry, School of Medicine, Zahedan University of Medical Sciences, Zahedan 9816743463, Iran; 2Department of Physiology, Faculty of Medical Sciences, Tarbiat Modares University, Tehran 1411713116, Iran; 3School of Medicine, Zahedan University of Medical Science, Zahedan 9816743463, Iran; 4Faculty of Advanced Medical Sciences, Department of Medical Biotechnology, Tabriz University of Medical Sciences, Tabriz 5166616471, Iran; 5Children and Adolescent Health Research Center, Resistant Tuberculosis Institute, Zahedan University of Medical Sciences, Zahedan 9816743463, Iran; 6Department of Cardiology, Cardiovascular Research Center, School of Medicine, Hormozgan University of Medical Science, Bandar Abbas 7916613885, Iran; 7Tehran Heart Center, Tehran University of Medical Science, Tehran 1416634793, Iran; 8Genetics of Non-Communicable Disease Research Center, Zahedan University of Medical Sciences, Zahedan 9816743463, Iran; 9Department of Genetics, School of Medicine, Zahedan University of Medical Sciences, Zahedan 9816743463, Iran; 10Research Institute of Oncology and Hematology, Cancer Care Manitoba-University of Manitoba, Winnipeg, MB R3E 0V9, Canada; 11Biology of Breathing Theme, Children Hospital Research Institute of Manitoba, University of Manitoba, Winnipeg, MB R3E 0V9, Canada; 12Faculty of Medicine in Zabrze, University of Technology in Katowice, 41-800 Zabrze, Poland; 13Department of Human Anatomy and Cell Science, Rady Faculty of Health Sciences, Max Rady College of Medicine, University of Manitoba, Winnipeg, MB R3E 0J9, Canada

**Keywords:** RTN4, rs34917480, polymorphism, breast cancer

## Abstract

Breast cancer (BC) is a complex disease caused by molecular events that disrupt cellular survival and death. Discovering novel biomarkers is still required to better understand and treat BC. The reticulon-4 (RTN4) gene, encoding Nogo proteins, plays a critical role in apoptosis and cancer development, with genetic variations affecting its function. We investigated the rs34917480 in RTN4 and its association with BC risk in an Iranian population sample. We also predicted the rs34917480 effect on RTN4 mRNA structure and explored the RTN4’s protein–protein interaction network (PPIN) and related pathways. In this case–control study, 437 women (212 BC and 225 healthy) were recruited. The rs34917480 was genotyped using AS-PCR, mRNA secondary structure was predicted with RNAfold, and PPIN was constructed using the STRING database. Our findings revealed that this variant was associated with a decreased risk of BC in heterozygous (*p* = 0.012), dominant (*p* = 0.015), over-dominant (*p* = 0.017), and allelic (*p* = 0.035) models. Our prediction model showed that this variant could modify RTN4’s mRNA thermodynamics and potentially its translation. RTN4’s PPIN also revealed a strong association with apoptosis regulation and key signaling pathways highly implicated in BC. Consequently, our findings, for the first time, demonstrate that rs34917480 could be a protective factor against BC in our cohort, probably via preceding mechanisms.

## 1. Introduction

Breast cancer (BC) is one of the most common malignancies globally and remains a significant cause of death and morbidity despite considerable improvement in diagnostic and therapeutic approaches [1,2,3,4]. A significant part of BC’s mortality and burden is due to our limited knowledge about underlying etiopathogenesis and predisposing factors, which is essential to develop new biomarkers to target the disease early and more efficiently [5]. BC originates from distorted molecular events that alter cellular proliferation, survival, and death [5,6]. These distorted events stem from genetic defects caused by interactions between numerous factors, including hereditary, environmental, reproductive, and lifestyle factors [7]. So far, several markers have been identified to confer a vital role in susceptibility, treatment response, and outcomes of BC (e.g., BRCA1/2, TP53, STK11, CD1, and/or PTEN) [8,9]. Still, a considerable part of BC’s molecular and genetic background remains to be elucidated to develop effective diagnostic, prognostic, and therapeutic tools containing BC’s mortality and burden.

Reticulon-4 (RTN4) is a ubiquitous protein-coding gene located on 2q16, encoding neurite outgrowth inhibitor (Nogo) proteins via several transcript variants (isoforms) as a result of different splicing and the utilization of the promoter region [10,11]. Nogo isoforms have specific tissue expression patterns, functions, and interactions in neural and non-neural cells [11,12]. The primary function of RTN4/Nogo isoforms in the cell is modulating endoplasmic reticulum morphology, while these proteins also contribute to cell proliferation, differentiation, and survival under certain circumstances [13,14]. Emerging evidence has demonstrated that these proteins regulate apoptosis and intracellular signaling pathways, particularly in cancer cells [15,16,17]. Briefly, Nogo proteins are suggested to be pro-apoptotic, and their overexpression can enhance apoptosis in cancer cells via several mechanisms [16], while their down-regulated expression has also been reported in several cancers, suggesting a tumor-suppressing role in certain types of cancers [18,19]. On top of that, RTN4/Nogo is also suggested to be pro-oncogenic by modulating signaling pathways critically involved in cancer development, such as PI3K/AKT and MAPK/ERK. Through constitutional RAS membrane localization and activation, Nogo proteins seem to trigger the PI3/AKT and MAPK/ERK signaling transduction pathways, rendering the cell incapable of programmed cell death while promoting cell proliferation, differentiation, and survival [20,21].

On the other hand, genetic variations in RTN4 may change its expression pattern and result in pathologic events associated with cancer [22,23]. As such a variation, rs34917480 is a three base-pair (CAA) Ins/Del polymorphism at the RTN4’s 3′ untranslated region (UTR) [24], a critical region in for regulation of gene expression [25]. Generally, polymorphisms of 3′UTR can lead to altered gene expression or abnormal protein production and be a predisposing factor for carcinogenesis [26]. rs34917480 has been associated with several human diseases, such as schizophrenia [24], cardiac diseases like dilated cardiomyopathy [27] and congenital heart diseases [28], uterine leiomyomas [29], as well as cancers like cervical squamous cell carcinoma [30], non-small-cell lung cancer [31], clear-cell renal-cell carcinoma [32], and hepatocellular carcinoma [33]. However, to our knowledge, no study has investigated the association of rs34917480 with breast cancer susceptibility.

We, therefore, conducted this population-based case-control study to investigate the association between RTN4 rs34917480 and susceptibility to BC in a sample population from the southeast of Iran. Further, we analyzed the effect of rs34917480 on the mRNA structure of RTN4 through bioinformatics analysis. We also explored RTN4’s interactions, functions, and implications in apoptosis and breast cancer development via predicting protein–protein interactions and enriched networks. This study’s results could help us better understand the molecular mechanisms, regulatory networks, and potential biomarkers associated with BC that can be considered in future investigations to develop target-specific diagnostic and therapeutic options.

## 2. Materials and Methods

### 2.1. Subjects

We recruited 437 individuals, comprising 212 female patients with BC as the case group and 225 age-matched healthy women as controls. All subjects were women living in southeast Iran. The inclusion criteria for the case group were women with breast cancer referred to a university-affiliated hospital (Ali-Ebn-e-Abitaleb Hospital, Zahedan, Iran) with confirmed malignant pathology, while women with other underlying medical conditions and benign breast pathologies were excluded. The control groups comprised healthy women recruited through routine checkup programs, while those with systemic diseases, cancer, lumps, or pathologies on breast examinations were excluded.

All the participants gave written informed consent for their involvement in this experiment. This study was conducted with approval from the committee of the review board of Zahedan University of Medical Sciences (IR.ZAUMS.REC.1397.225).

### 2.2. DNA Extraction and Genotyping

For the genotyping of the rs34917480 CAA Ins/Del polymorphism of RTN4, specific primers were designed via the allele-specific PCR (AS-PCR) method, and the protocol for the current study was as follows:

The whole blood samples of patients and controls were collected in tubes containing EDTA and stored at −20 °C until further use and moved to 4 °C before DNA extraction. Genomic DNA from whole blood was extracted using the salting-out method as described previously [34,35]. The primers sequences for the AS-PCR were:-Deletion allele (F1): CAA-F 5′-GTCTGTGCAATGAAATTGATGTTGGA-3′;-Insertion allele (F2): CAA-F 5′-GTCTGTGCAATGAAATTGATGTTGTT-3′;-Generic reverse primer: 5′ACCGGTAAAGCAGGAATGACAA-3′.

For this reaction, two allele-specific forward primers were designed—one for a deletion allele (F1) and the other for an insertion allele (F2)—along with one consensus generic reverse primer. Upon the completion of PCR, two equally long bands were produced, with an amplicon size of 308 bp for both insertion and deletion—the presence of each band represents the presence of the corresponding allele [35,36]. The forward and reverse primers of pri-miR-34b/c (5′-CCTCTGGGAACCTTCTTTGACCTGT-3′ and 5′-CCTGGGCCTTCTAGTCAAATAGTGA-3′, respectively) were used as the internal control and added to each tube to ensure successful amplification, which was indicated by the presence of 212 bp bands for the internal control in all samples.

PCR was performed in 25 μL reaction volumes containing 1 μM of each primer, 250 μM of each dNTP, 1 U Taq DNA polymerase with 1.5 mM MgCl2, and ~100 ng of genomic DNA. The PCR cycling conditions were as follows: initial denaturation at 95 °C for 5 min followed by 30 cycles of 30 s at 95 °C, annealing at 59 °C for 30 s, and extension at 72 °C for 30 s. The final extension step was performed at 72 °C for 5 min. The PCR products were separated via electrophoresis in 2% agarose gels and were observed after staining with ethidium bromide (Figure 1). To ensure quality assessment, we randomly re-genotyped 10% of the samples to show the reproducibility of the genotyping results. Additionally, to confirm the AS-PCR genotyping results, we randomly selected DNA samples from AS-PCR amplification for DNA sequencing (n = 4 for each genotype; see the Results section and Appendix A).

### 2.3. Computational Analyses

#### 2.3.1. mRNA Secondary Structure Prediction

We examined whether the RTN4 rs34917480 polymorphism can affect the secondary structure as well as the local folding of RTN4 mRNA using RNAfold, a web-based tool to predict the RNA secondary structure via free energy minimization (i.e., minimal free energy (MFE) structure) [37,38]. The MFE is defined as “the secondary structure that contributes a minimum of free energy” [39]. A 500 bp region flanking the variant was analyzed to ensure sufficient sequence context that could be influenced by the variant, potentially modulating RTN4 gene expression or mRNA structure. This analysis showed us the estimated influence of rs34917480 on the folding, stability, or other aspects of the mRNA structure, potentially impacting the gene expression or functional properties of the RTN4 gene by estimated changes in the free energy of the mRNA’s thermodynamic ensembles.

#### 2.3.2. Protein–Protein Interaction Network

We constructed a protein–protein interaction network (PPIN) using the STRING database (v11.5) [40], according to our previous experience [41], to illustrate the potential interacting proteins with RTN4 in separate functional clusters based on existing evidence. The statistics and details that are used to construct the PPIN are provided in Table 1, in which nodes represent proteins or protein entities; edges—the interaction or connection between nodes; the expected number of edges—expected edges based on a random model or null hypothesis; the average node degree—the average number of connections each node has; Avg. local clustering coefficient—the tendency of neighboring nodes to form clusters; inflation parameter—the parameter used in the Markov Cluster Algorithm (MCL) for clustering analysis that influences the granularity and size of the resulting clusters; and enrichment *p*-value, revealing the statistical significance of the observed clustering (see Appendix A for clusters detail).

### 2.4. Statistical Analysis

All statistical analyses were performed using the statistical software SPSS version 22 (IBM Corp., Armonk, NY, USA). The genotype frequency of Ins/Del polymorphisms in the RTN4 gene was obtained using directed counting, and deviation from Hardy–Weinberg equilibrium (HWE) was assessed using the chi-square test. Logistic regression was used to analyze the relationship between the RTN4 3-bp Ins/Del variant and BC susceptibility. The odds ratio (95% CI) was used to examine the effect of differences between alleles. A *p*-value less than 0.05 was considered statistically significant.

## 3. Results

### 3.1. Genotype and Allele Frequency

The present population-based case–control study consisted of 212 BC patients with an average age of 48.34 ± 10.82 years and 225 cancer-free women with a mean age of 49.30 ± 11.96 years. The age was not statistically different between the groups (*p* = 0.92; Table 2).

Also, the HWE calculation for patients and controls of each polymorphism showed that the distributions of RTN4 rs34917480 in patients, but not controls, were significantly deviated from HWE (*p* < 0.0001, χ^2^ = 16.15 and *p* = 0.81, χ^2^ = 0.05, respectively, Table 3).

Table 4 displays the genotypes and allele frequencies of the RTN4 rs34917480 polymorphism in BC cases and controls. The results showed that the Ins/Del polymorphism is associated with a lower risk of BC in heterozygous (OR = 0.59, 95% CI = 0.39–0.89, *p* = 0.012 Ins/Del vs. Del/Del), dominant (OR = 0.62, 95% CI = 0.43–0.91, *p* = 0.015, Ins/Del + Ins/Ins vs. Del/Del), and over-dominant (OR = 0.62, 95% CI = 041–0.92, *p* = 0.017, Ins/Del vs. Del/Del + Ins/Ins) models and insertion alleles (OR = 0.72, 95% CI = 0.52–0.99, *p* = 0.035, Ins vs. Del). But there were no significant differences in homozygous (OR = 0.74, 95% CI = 0.38–1.46, *p* = 0.388, Ins/Ins vs. Del/Del) or recessive (OR = 0.92, 95% CI = 0.47–1.78, *p* = 0.800, Ins/Ins vs. Ins/Ins + Del/Del) models.

We randomly re-genotyped 10% of the samples as part of our quality control process for genotyping, and the results demonstrated 100% reproducibility and no errors in the findings. Also, selected amplified DNA products of each genotype (n = 4 for each genotype) were examined via DNA sequencing, which further confirmed our AS-PCR results (see Appendix A for sequencing results).

Further, we examined the association between RTN4 rs34917480 and clinicopathological features of BC, including the age and tumor size at the time of diagnosis, histological subtype, grade and stage, and the status of estrogen receptor (ER), progesterone receptor (PR), and HER2 receptors of patients. As Table 5 depicts, there was no significant association between this variant and clinicopathological features in our study, except for PR status, which shows a significantly higher prevalence of the PR-positive DD genotype in BC patients (*p* = 0.031, Table 5).

### 3.2. Computational Findings

#### 3.2.1. mRNA Second Structure Prediction

The 500 bp region flanking the variant was used to evaluate the influence of rs34917480 (3 bp Ins/Del) on the RTN4 mRNA structure. The putative influence of the rs34917480 variant on the local folding structure changes in RTN4 is shown in Figure 2A. The free energy of the thermodynamic ensemble for insertion and deletion was −252.68 and −251.85 kcal/mol, respectively, which resulted in higher entropy in the specific regions shown in Figure 2. This higher entropy, in turn, reflects the higher variability in RNA conformations within the thermodynamic ensemble and is associated with increased instability. The positional entropy for each position is shown in Figure 2B. The result demonstrated that the rs34917480 variant causes changes in the MFE structure, the thermodynamic ensemble of RNA structures, and the centroid structure (an alternative way of predicting RNA secondary structure, compared to MFE, which has the minimum total base-pair distance to all sets of secondary structures) [38]. These estimated changes can affect the secondary structure of mRNA due to making different thermodynamic ensembles more or less favorable, thereby leading to potential alterations in mRNA structure, function, and translational activities [42].

#### 3.2.2. Protein–Protein Interaction Network (PPIN)

The constructed network via STRING shows the functional connection between nodes with the highest degree and betweenness (minimum interaction score = 0.7 (high confidence)) in four different clusters (see Figure 3’s legend for the clustering statistics and Appendix A for cluster details). Additionally, we selected the top functional enrichment of pathways retrieved from the PPIN, which could be more critical in BC pathophysiology, as summarized in Table 6, in which observed refers to the number of proteins that the RTN4 PPIN network is associated with, background refers to the total number of proteins in the PPIN network that RTN4 can potentially be associated with, and strength represents the significance of the association between the proteins in the PPIN network and the specific functional category (full details of each network are provided in Appendix A).

Although RTN4 significantly contributes to nervous system development, our predicted network suggests that RTN4/Nogo proteins are strongly implicated in cellular growth, development, and survival pathways (Table 4). Nevertheless, plenty of nodes are contributing to RTN4 PPIN for regulating the mentioned processes (Figure 3). Among them, we identified the following nodes as hub-driver nodes based on key associating factors connected to BC carcinogenesis: RTN3 (reticulon-3), RTN4IP1 (reticulon-4-interacting protein 1), NUS1 (nuclear undecaprenyl pyrophosphate synthase 1), S1PR2 (sphingosine-1-phosphate receptor 2), NGF (nerve growth factor), NTRK1 (neurotrophic tyrosine kinase receptor type 1), NTRK2 (neurotrophic tyrosine kinase receptor type 2), NGFR (neural growth factor receptor), SORT1 (sortilin 1), ARHGDIA (Rho GDP-dissociation inhibitor A), APP (amyloid precursor protein), and BACE1 (β-site amyloid precursor protein cleaving enzyme-1). There is considerable evidence that links these nodes with breast tumor cell development and survival by exhibiting anti-apoptotic and/or modulatory signaling pathway activity, implying a significant connection of RTN4 PPIN with BC development.

## 4. Discussion

In the present study, using a sample from the southeast of Iran, we found that the 3-bp CAA Ins/Del polymorphism in the 3′UTR of RTN4 significantly decreases the BC risk and relates to progesterone receptor status. We also discovered that rs34917480 alters the free energy of the mRNA folding and, subsequently, mRNA function and translation, resulting in a potentially different product with potentially different activities. Finally, we further elucidated at the molecular level that not only is RTN4/Nogo involved in the regulation of apoptosis, but it is also strongly associated with several key signaling pathways implicated in BC carcinogenesis.

The reticulon gene family includes four genes—RTN1, RTN2, RTN3, and RTN4/Nogo—with nine alternative transcripts, some of which have unknown functions [11]. As a member, RTN4/Nogo is a 75 kb gene on chromosome 2 (2p16), including 14 exons and 8 introns [13]. RTN4 encodes neurite growth inhibitor proteins (Nogo) through several transcript variants, including three significant isoforms (Nogo A-B-C) [11,13]. Nogo-A is the largest isoform, known as a neuronal regeneration inhibitor; Nogo-B has broader expression in different tissues and is mostly involved in vascular remodeling, inflammatory processes, as well as tissue repair; and lastly, Nogo-C is highly expressed in skeletal muscle with still barely known functions [10,11]. Growing evidence has linked these isoforms with critical interactions in vital cellular systems such as cellular proliferation, survival, and apoptosis, suggesting oncogenic roles for RTN4/Nogo [13,43]. The oncogenic roles of RTN4/Nogo have been investigated in various cancers, including oligodendroglioma [44]; non-small-cell lung cancer [45]; hepatocellular [46], cervical [47], colorectal [48], and nasopharyngeal carcinoma [49]; as well as BC [50,51,52].

BC is a heterogenous disease, including multiple tumor entities with similar manifestations, each characterized by distinct molecular, morphological, and clinical hallmarks [53]. The molecular portrait of BCs is based on the expression of hormone receptors (ER and PR, HER2 (or ErbB-2), or none: basal-like or triple-negative breast cancers (TNBCs)) [53,54]. These subtypes illustrate some underlying molecular signatures of the disease; ER and PR overexpression lead to cell cycle progression via the estrogen/ER pathway, while in HER2-positive breast tumors, HER2 gene amplification or protein overexpression activates the PI3k/AKT/mTOR and RAS/RAF/MAPK pathways, stimulating cell proliferation, differentiation, and survival [54,55,56].

In line with our results from the predicted network, several studies suggested that RTN4/Nogo can modulate PI3K/Akt and MAPK/ERK activation and be associated with cancer development [20,21,22]. In studies investigating the functional role of Nogo-B and Nogo-C, these proteins are suggested to influence RAS membrane recruitment, phosphorylation, and activation by binding to their specific receptors [20,21]. RAS oncoproteins are mainly involved in the PI3/AKT and MAPK/ERK pathways, and their over-activity has been detected in several human cancers, especially BC [57]. Thus, Nogo proteins may recapitulate the oncogenic function of downstream AKT and ERK1/2 pathways in BC [20,21]. In this line, studies have suggested the Nogo-B receptor (NgBR) as a critical element for Ras activation in breast tumor cells and have reported significant association between the high expression of Nogo-B/NgBR and ER-positive/HER2-negative BC [51,58]. These results are consistent with other researchers suggesting the increased expression of Nogo-B and/or NgBR in BC on top of an inverse correlation of Nogo-B expression with survival and treatment response to tamoxifen and paclitaxel, signifying the prognostic and therapeutic potential of RTN4 in BC [22,52]. In addition to these suggested molecular bases of RTN4/Nogo in BC carcinogenesis, our predicted PPI network might indicate several other genes, proteins, and signaling pathways associated with BC that are less commonly mentioned in RTN4 studies, such as NUS1, S1PR2, NGF/NTRK1-2, BACE1, and APP.

NUS1 (nuclear undecaprenyl pyrophosphate synthase 1) encodes the Nogo B receptor (NgBR), a specific receptor for Nogo-B [59]. The NogoB/NgBR signaling axis is suggested to modulate the EGF/Ras/Raf/ERK and PI3K/Akt signaling pathways [60] and increases the expression of an anti-apoptotic protein called survivin, both of which are shown to be associated with the development and survival of breast tumor cells [51]. S1PR2 (sphingosine-1-phosphate receptor 2) is a G-protein coupled receptor for S1P (sphingosine-1-phosphate), a bioactive sphingolipid metabolite produced by SphK1 (sphingosine kinase1), both of which have been suggested to be associated with BC carcinogenesis [61]. Once produced by SphK1, S1P binds to S1PRs and EGFRs, initiating several oncogenic signaling pathways implicated in BC, such as the NF-kB, STAT3 PI3K/AKT, Ras/Erk, Rak1/PLC, EGFR (HER2), and ERK1/2 pathways [62]. S1PR2’s involvement in BC has already been reported [63], but the exact mechanism and significance of S1PR2 in BC carcinogenesis still need to be determined. Further, the NGF/NTRK1 (or TrkA) signaling axis modulates the survival, growth, and prevention of apoptosis through the PLC*γ*1, the PI3K-Akt, and the Ras/Raf/MEK/ERK 1/2 pathways, which are implicated in BC carcinogenesis [64,65].

The activation of the NGF/NTRK2 (or TrkB) axis also modulates several signaling pathways related to survival and cell growth, including PI3K-AKT, NF-kB, Jak/STAT, and VEGF pathways in response to neurotrophins, such as BDNF (brain-derived neurotrophic factor) [66,67]. In BC, these pathways are aberrantly active and are suggested to be linked with poor clinical prognosis and reduced survival [66,68]. Moreover, BACE1 (β-site amyloid precursor protein cleaving enzyme-1) is an aspartic protease that cleaves APP to produce β-amyloid [69]. It has been suggested that BACE1 could alter cancer progression by changing the tumor microenvironment, activating STAT3 signaling via IL-6R, and subsequently maintaining tumor-promoting macrophages [70]. In BC, low levels of BACE1 are shown to be associated with cancer development [71]. Lastly, APP (amyloid precursor protein) is a transmembrane protein that participates in adhesion, axonogenesis, and neurite growth in neural tissue [72]. However, in non-neural tissues, APP has been suggested to play a central role in growth and angiogenesis, while APP overexpression has been shown in several cancers, implying its role in tumor proliferation [73]. In BC cells, APP overexpression is associated with proliferation, motility, and invasiveness, presumably through AKT/FOXO, AKT/GSK3-β, and MAPK signaling pathways [74].

Alternatively, the pro-apoptotic role of RTN4/Nogo has also been suggested to be associated with cancer development [16,18]. Evidence has shown that the overexpression of Nogo proteins can induce apoptosis in tumor cells, suggesting a tumor suppressor effect, while normal cells seem resistant to this Nogo-dependent apoptosis [16]. Conversely, their decreased expression in cancer cells might lead to apoptosis evasion [15,16,17,18]. Dysregulation between pro-apoptotic and anti-apoptotic proteins is a fundamental element in tumorigenesis, and therefore, any distortions in the expression of RTN4/Nogo, as pro-apoptotic proteins, may increase the susceptibility to cancer, including BC [75,76]. Several mechanisms for the pro-apoptotic activity of Nogo isoforms have been so far described: inducing ER stress and signaling pathways, expressing survivin, interacting with anti-apoptotic proteins such as the Bcl-xL-Bcl-2 family and c-FLIP, and interacting with the JNK-c-Jun MAPK signaling pathways [15,16,17,18,51]. In harmony, our predicted PPI network indicates that RTN4/Nogo isoforms might also dysregulate apoptotic activity via distinctive interactions less mentioned in RTN4 studies, such as RTN3, NGFR, SORT1, ARHGDIA, and RAC1.

RTN3 is a protein-coding gene suggested to modulate Bcl-2 localization and anti-apoptotic activity (the same mechanism suggested for RTN4B/Nogo B), indicating their essential role in cell survival [77]. Further, NGFR—also called p75NTR—is a transmembrane receptor belonging to the TNF (tumor necrosis factor) family expressed in many tissues [78]. Depending on the cancer type, it has been reported that NGFR can act as an oncosuppressor and an oncogene [79,80]. The activation of the NGF/NGFR signaling axis activates transcription factors (depending on the cell context) and induces apoptosis (via the JNK cascade) or cell survival (via the NF-*κ*B pathway) [65]. In addition, precursors of NGF (called proNGF) bind to NGFR-sortilin (or SORT1), which can induce cell death [81]. In BC, NGF, proNGF, and their receptors are shown to be expressed in tumor cells and stimulate proliferation, migration, and tumor cell survival through these distinct signaling pathways mentioned above [82,83]. Moreover, in this cluster, ARHGDIA (Rho GDP-dissociation inhibitor A) inhibits GDP dissociation from Rho GTPase proteins, including RAC1, CDC42, and RHOA, holding these proteins in an inactivated state [84]. Rho GTPases, especially RAC1, increase anti-apoptotic activity, angiogenesis, proliferation, and metastasis [85]. RAC1 plays an anti-apoptotic role in BC through NF-κB signaling pathways and increased Bcl-2 and Bcl-x levels [86,87]. Additionally, the down-regulation of ARHGDIA has been shown in BC [88]. However, depending on tumor type and stage, ARHGDIA can act as an oncosuppressor or oncogene [89].

Taken together, this evidence may enlighten several oncogenic aspects of RTN4/Nogo in BC development. Apart from RTN4’s suggested oncogenic roles, polymorphisms in RTN4 may influence its expression and function, leading to pathologic conditions like cancer [23]. Among these polymorphisms, increasing studies have focused on the insertion/deletion of TATC (rs71682890) and CAA (rs34917480) in 3′UTR of RTN4 [90]. The 3′UTR is considered a transcriptional regulator of a gene, modifying gene expression, while variations in this critical regulatory region are associated with several diseases and cancers [25,91]. Similarly, it is suggested that rs34917480 can dysregulate the RTN4/Nogo isoform’s expression [90], leading to several diseases, particularly cancers.

Among the first studies investigating rs34917480 association with cancers, Shi et al. reported that this polymorphism is associated with cervical squamous cell carcinoma risk and higher clinical stages in the Chinese population, suggesting a prognostic value of rs34917380 [30]. Another case–control study on the Chinese population by Lu et al. revealed that the CAA deletion allele in the dominant model and Ins/Del genotype in the codominant and over-dominant model increases the risk of non-small-cell lung cancer (NSCLC) significantly in 411 patients [30,31]. These results were consistent with the study by Wang et al., demonstrating that the deletion allele and Ins/Del and Del/Del genotypes of rs34917480 were associated with increased HCC risk [31,33]. However, Pu et al. and Shi et al. found no significant association between CAA polymorphism with clear cell renal carcinoma and cervical squamous cell carcinoma, respectively, although they both reported significant differences for TATC (rs71682890) polymorphism [30,32].

Contrary to other researchers, here, we revealed that the insertion allele (CAA) of this polymorphism is significantly associated with the decreased risk of BC (OR = 0.72) as in heterozygous (Ins/Del vs. Del/Del), dominant (Ins/Del + Ins/Ins vs. Del/Del), over-dominant (Del/Ins vs. Del/Del-Ins/Ins), and codominant (Ins vs. del) genetic models. We also found that rs34917480 significantly correlates with progesterone receptor status, consistent with other researchers stressing the prognostic value of rs34917480. Additionally, we conducted PPIN and function analysis for RTN4 to justify potential contributors to the observed effect of rs34917480 of RTN4 in BC. As a result, we demonstrated that several nodes in RTN4 PPIN have been associated with BC carcinogenesis via several distinct mechanisms (e.g., NUS1, S1PR2, proNGF/NGF/NGFRs, ARHGDIA, APP, and BACE1, as depicted in Figure 3 and pathways shown in Table 6), through which, we assume RTN4 can lead to BC development. Lastly, we predicted the effect of rs34917480 on the secondary structure of RTN mRNA and found that this variant induces changes in the MFE structure, the thermodynamic ensemble, and the centroid structure of mRNA (the positional entropy for each position shown in a Figure 2B), subsequently leading to altered mRNA function and translation.

Considering our findings and the abovementioned mechanisms, we assume this polymorphism might be associated with higher RTN4/Nogo isoforms, highlighting their pro-apoptotic activity that suppresses tumor development. Notably, we assume that this tumor suppressor role (suggested to be intensified by this polymorphism) might reduce or overcome the oncogenic role of RTN4/Nogo isoforms induced by PI3K/AKT and MAPK/ERK activation. Further studies are needed to elucidate the observed tumor suppressor effect of rs34917480.

Nevertheless, our study faced some limitations that should be considered in the interpretation of our result and also in further studies. Firstly, the number of subjects was relatively small. Moreover, we have only investigated one variant, which, given that numerous variants may affect a gene’s function, seems insufficient to reach conclusions with. Future research should focus on larger sample sizes, diverse ethnicities, and other polymorphisms. There is ample scope for future research on expression levels of RTN4 polymorphism in BC patients and investigating the precise cellular mechanisms involved. Therefore, further experimental studies should consider these results to reveal the interaction.

## 5. Conclusions

The present study suggests a significant association of the 3-bp CAA Ins/Del polymorphism in 3′UTR of RTN4 with BC susceptibility and progesterone receptor status in a southern Iranian population. Furthermore, we showed that the variant could potentially affect the secondary structure of the mRNA through bioinformatics computations. With a focus on carcinogenesis, we draw a PPIN showing the corresponding genes that shape the RTN4 functionality within a network and can be considered in future investigations to develop target-specific diagnostic and therapeutic options for BC.

## Figures and Tables

**Figure 1 life-13-01549-f001:**
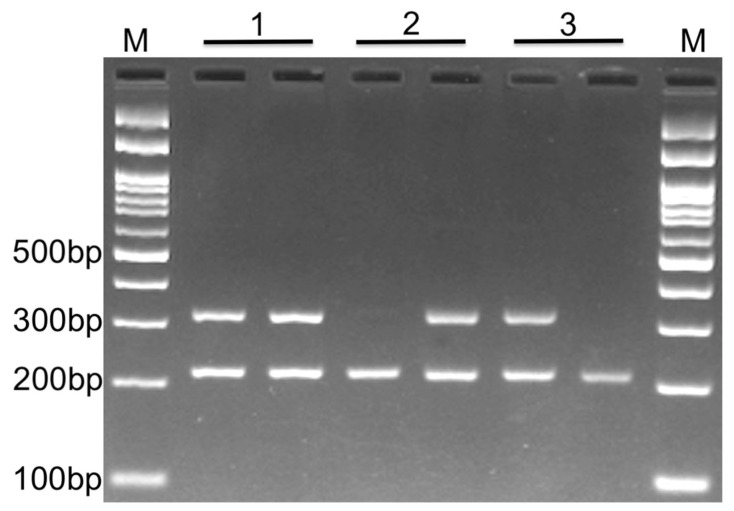
Gel electrophoresis. Photograph of electrophoresis pattern of the allele-specific PCR to detect RTN4 rs34917480. Each sample has two tubes containing either an insertion or deletion primer (along with reverse and internal control primers in both tubes). Lane 1: Insertion/deletion, which is the result of the existence of both alleles for that sample (heterozygous); Lane 2: Deletion/deletion, which depicts the presence of only a deletion allele for the sample (homozygous); Lane 3: Insertion/insertion, which depicts the presence of only an insertion allele for the sample (homozygous); product size for insertion or deletion and internal control is 308 bp and 212 bp, respectively. M: DNA marker.

**Figure 2 life-13-01549-f002:**
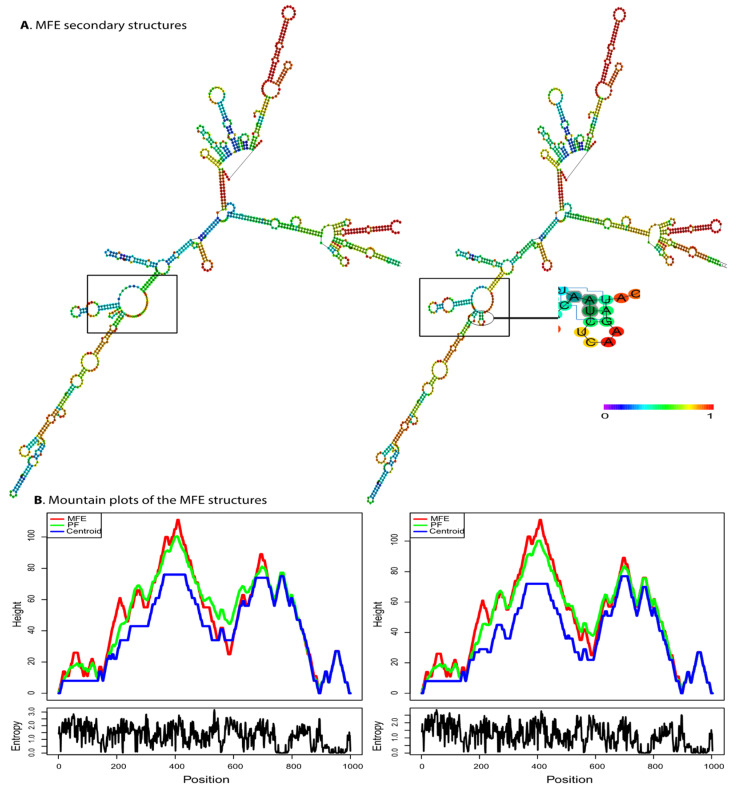
Predicted secondary structure of RTN4 mRNA in the presence of rs34917480 via RNAfold. (**A**) The effect of the rs34917480 variant on the structure of RTN4 (left, wild type, and right, mutant) based on minimum free energy (MFE) of nucleotide base pairing (see the color gradient). The free energy of the thermodynamic ensemble for insertion and deletion was −252.68 and −251.85 kcal/mol, respectively. Higher entropy is associated with increased instability. (**B**) The mounting plot for MFE, thermodynamic ensemble (PF), and centroid prediction for structures depicted in panel A shows changes caused by the rs34917480 variant. The slopes, plateaus, and peaks show helices, loops, and hairpins. The lower sections show the entropy of structures.

**Figure 3 life-13-01549-f003:**
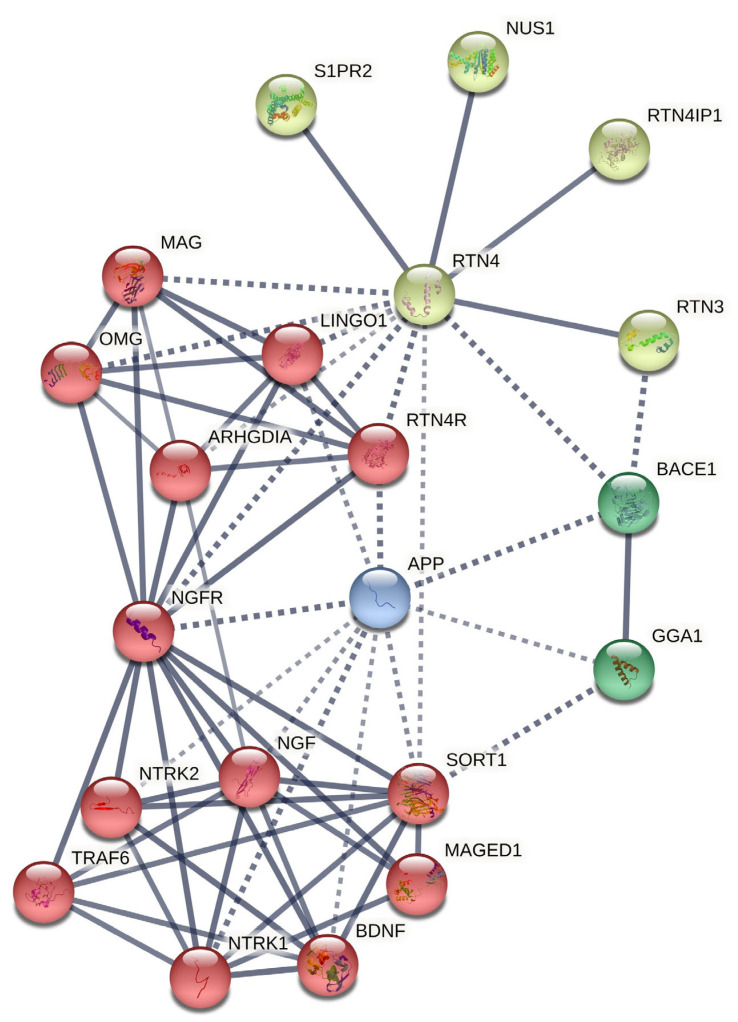
RTN PPIN. Four different clusters of genes are shown in this network, based on MCL clustering (inflation parameter = 6.0). Dotted lines show edges between clusters. The number of nodes = 21; the number of edges = 65; average node degree = 6.19; average local clustering coefficient = 0.781; expected number of edges = 22; PPIN enrichment *p*-value = 5.06 × 10^−14^; Edge Confidence in the network = high (0.700); line thickness indicates the strength of data support.

**Table 1 life-13-01549-t001:** PPIN network properties and statistics.

Description	Nodes	Edges	Expected Number of Edges	Avg. Node Degree	Avg. Local Clustering Coefficient	Inflation Parameter (MCL)	Enrichment *p*-Value
RTN4 PPIN	21	65	22	6.19	0.781	6	5.06 × 10^−14^

**Table 2 life-13-01549-t002:** Demographic characteristics of patients and control.

Characteristics		Patients, n (SD)	Control, n (SD)	*p **
Age	Mean (SD)	48.34 (10.82)	49.30 (11.96)	0.92
	Median (IQR)	48.00 (16.0)	46.0 (17.0)	

* Mann–Whitney test. SD, standard deviation; IQR, interquartile range.

**Table 3 life-13-01549-t003:** Allele and genotype frequency of RTN4 rs34917480 in patients and controls.

Polymorphism	Group	Genotypes, n (%)			Allele, n (%)		HWE	
		Del/Del	Ins/Del	Ins/Ins	Del	Ins	χ^2^	*p*
rs34917480	Patients	134 (63.2)	60 (28.3)	18 (8.5)	328 (77.4)	96 (22.6)	16.15	<0.0001
	Controls	116 (51.6)	88 (39.1)	21 (9.3)	320 (71.1)	130 (28.9)	0.05	0.81

HWE, Hardy–Weinberg equilibrium, Del, deletion, Ins, insertion.

**Table 4 life-13-01549-t004:** Association between genotypes and alleles of RTN4 rs34917480 and risk of breast cancer.

Polymorphisms	Genetic Models	Patients, n (%)	Control, n (%)	OR (95% CI) *	*p* **
RTN4 rs34917480	Codominant				
	Del/Del	134 (63.2)	116 (51.6)	1.00	-
	Ins/Del	60 (28.3)	88 (39.1)	0.59 (0.39–0.89)	0.012
	Ins/Ins	18 (8.5)	21 (9.3)	0.74 (0.378–1.46)	0.388
	Dominant				
	Del/Del	134 (63.2)	116 (51.6)	1.00	-
	Ins/Del + Ins/Ins	78 (36.8)	109 (48.4)	0.62 (0.43–0.91)	0.015
	Recessive				
	Del/Del + Ins/Del	194 (91.5)	204 (90.7)	1.00	-
	Ins/Ins	18 (8.5)	21 (9.3)	0.92 (0.47–1.78)	0.8
	Over-dominant				
	Del/Del + Ins/Ins	152 (71.7)	137 (60.9)	1.00	-
	Ins/Del	60 (28.3)	88 (39.1)	0.62 (0.41–0.92)	0.017
	Alleles				
	Del	328 (77.4)	320 (71.1)	1.00	-
	Ins	96 (22.6)	130 (28.9)	0.72 (0.53–0.99)	0.035

Del, deletion, Ins, insertion; * Adjusted for age; ** Fisher’s exact.

**Table 5 life-13-01549-t005:** Association of *RTN4* gene polymorphism with clinicopathological characteristics of breast cancer patients.

Characteristic of Patients	Genotypes			*p*
	Del/Del n (%)	Ins/Del n (%)	Ins/Ins n (%)	
Age, years				0.32
≤50	86 (67.2)	32 (25.0)	10 (7.8)	
>50	48 (57.1)	28 (33.30	8 (9.5)	
Tumor size, cm				0.58
≤2	41 (59.9)	23 (33.3)	5 (7.2)	
>2	90 (66.2)	36 (26.5)	10 (7.4)	
Histology				0.085
Adenocarcinoma	22 (54.2)	19 (45.2)	1 (2.4)	
Ductal carcinoma	94 (68.1)	33 (23.9)	11 (8.0)	
Lobular carcinoma	5 (83.3)	0 (0)	1 (16.7)	
Mucinous carcinoma	11 (61.1)	5 (27.8)	2 (11.1)	
Grade				0.094
I	23 (63.9)	6 (16.7)	7 (19.4)	
II	68 (63.6)	33 (30.8)	6 (5.6)	
III+IV	24 (66.7)	8 (22.2)	4 (11.1)	
Stage				0.88
I	22 (61.1)	12 (33.3)	2 (5.6)	
II	48 (62.3)	24 (31.2)	5 (6.5)	
III	40 (63.5)	16 (25.4)	7 911.1)	
IV	23 (67.6)	8 (23.5)	3 (8.8)	
Estrogen receptor status				0.16
Positive	84 (67.7)	27 (21.8)	13 (19.5)	
Negative	46 (62.2)	24 (32.4)	4 (5.4)	
Progesterone receptor status				0.031
Positive	84 (70.6)	23 (19.3)	12 (10.1)	
Negative	45 (57.7)	28 (35.9)	5 (6.4)	
HER2 status				0.66
Positive	68 (62.4)	33 (30.3)	8 (7.3)	
Negative	66 (66.0)	25 (25.0)	9 (9.0)	

**Table 6 life-13-01549-t006:** Summary of functional enrichment in the RTN4 network.

Description	Observed	Background	Strength	FDR *
Selected biological process (GO) ^†^				
Regulation of plasma membrane-bounded cell projection organization	13	687	1.25	9.31 × 10^−11^
Regulation of cell development	14	956	1.13	9.31 × 10^−11^
Neurotrophin trk receptor signaling pathway	6	20	2.45	2.89 × 10^−10^
Regulation of cell differentiation	16	1874	0.9	8.70 × 10^−10^
Regulation of multicellular organismal development	16	2096	0.85	3.49 × 10^−9^
Regulation of the developmental process	17	2648	0.78	5.33 × 10^−9^
Regulation of multicellular organismal process	17	3227	0.69	8.35 × 10^−8^
Regulation of anatomical structure morphogenesis	12	1095	1.01	9.42 × 10^−8^
Regulation of apoptotic process	13	1550	0.89	2.67 × 10^−7^
Negative regulation of cell differentiation	10	728	1.11	5.29 × 10^−7^
Positive regulation of cell development	9	556	1.18	9.80 × 10^−7^
Selected molecular function (GO) ^††^				
Nerve growth factor receptor activity	2	2	2.97	0.0035
Ganglioside gt1b binding	2	3	2.79	0.0051
Nerve growth factor receptor binding	2	5	2.57	0.0069
Tumor necrosis factor receptor superfamily binding	3	48	1.77	0.0069
Amide binding	5	369	1.1	0.0105
Signaling receptor binding	8	1581	0.67	0.0332
Cytokine receptor binding	4	264	1.15	0.0345
Protein binding	16	7026	0.33	0.0384
Selected KEGG pathway ^†††^				
MAPK signaling pathway	6	288	1.29	8.20 × 10^−5^
Ras signaling pathway	5	226	1.31	0.00043
PI3K-Akt signaling pathway	5	350	1.12	0.0026
Selected Reactome pathway ^††††^				
p75 NTR receptor-mediated signaling	10	96	1.99	5.08 × 10^−15^
Signal transduction	15	2741	0.71	1.65 × 10^−6^
Activation of TRKA receptors	3	6	2.67	2.76 × 10^−5^
p75NTR recruits signaling complexes	3	13	2.33	0.00016
NF-kB is activated and signals survival	3	13	2.33	0.00016
NRIF signals cell death from the nucleus	3	16	2.24	0.00022
Cell death signaling via NRAGE, NRIF, and NADE	4	75	1.7	0.00025

* False discovery rate (FDR) is used for multiple comparison correction. ^†^ See Appendix A for the complete table of biological processes. ^††^ See Appendix A for the complete table of molecular functions. ^†††^ See Appendix A for the complete table of the KEGG pathway. ^††††^ See Appendix A for the complete table of the Reactome pathway.

## Data Availability

The data presented in this study are available on request from the corresponding author.

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
