# Peer review of "The Reticulon-4 3-bp Deletion/Insertion Polymorphism Is Associated with Structural mRNA Changes and the Risk of Breast Cancer: A Population-Based Case–Control Study with Bioinformatics Analysis"

_life, 2023, doi:10.3390/life13071549_

Round 1
Reviewer 1 Report
The article "The Reticulon-4 3-bp Deletion/Insertion Polymorphism is Associated with mRNA Structural Changes and Risk of Breast Cancer: A Population-based Case-control Study with Bioinformatics Analysis" is an interesting study of RTN4 rs34917480 polymorphism associated with breast cancer risk in Iranian female population. The research is based on bioinformatics analysis. The article has a standard layout. Description of methods Authors can extend. The figures are clear and accurately visualize the obtained results. References the mostly contain publications from the last 10 years, so they are up to date. The article has a research aspect and potential application in the clinic. Many transcriptomic and proteomic studies indicate that this type of research can be an element of diagnostic tests and be part of, for example, onco-tests. In my opinion, the article "The Reticulon-4 3-bp Deletion/Insertion Polymorphism is Associated with mRNA Structural Changes and Risk of Breast Cancer: A Population-based Case-control Study with Bioinformatics Analysis" can be accepted for publication in LIFE in its present form.
Grammatical language form should be introduced.
Author Response
Response: The authors greatly appreciate the positive insight of the respected reviewer on our manuscript and highly appreciate the efforts in reviewing our manuscript.
Reviewer 2 Report
Although this is a case study for a specific population, which seems an interesting idea. However, the results are clearly not presented well. Extensive and careful editing is required to present the meaningful results providing supporting references. A few remarks ------------------What is PPIN in Abstract?
This information is too much to give in Abstract" Our findings revealed that this variant was associated with decreased risk of BC 57 in heterozygous (OR=0.59, 95%CI= 0.39-0.89, p=0.012 D/I vs. D/D), dominant (OR=0.62, 95%CI=0.43- 58 0.91, p=0.015, D/I + I/I vs. D/D), over-dominant (OR=0.62, 95%CI=041-0.92, P=0.017, D/I vs. D/D-I/I) 59 and allelic (OR=0.72, 95%CI=0.52-0.99, p=0.035, I vs. D) models, while there were no significant dif- 60 ferences in homozygous (OR=0.74, 95%CI=0.38-1.46, p=0.388, I/I vs D/D), or recessive (OR=0.92, 61 95%CI=0.47-1.78, p=0.800, I/I vs I/I + D/D) models " Can you generalize it by highlighting a few points rather than detailed comments in the abstract.. In keywords: Keywords: RTN4, rs34917480, polymorphism, 5 What is the meaning of 5? Can authors explain the details provided in Table 1? How is it relevant in this study? In Table 2: 49.30 (11.96) is SD of the age for control, and in text it is 11.97... Please be careful with all the numbers that authors report in text.. What is DD DI II D I in Table 3, 4, 5? Not clear when we even read the text.. "" The 500-bp region flanking the variant was used to evaluate the influence of the 227 rs34917480 (3-bp ins/del) on the RTN4 mRNA structure.. " What is the length of the variant? Please mention it.. It is not clear why mRNA prediction is necessary here? Please explain carefully.. Which tool did the authors use to convert genes to mRNA structure first? What is the meaning of this sentence? Our result demonstrated that the rs34917480 variant causes 231 changes in the MFE structure, the thermodynamic ensemble of RNA structures, and the 232 centroid structure. What is the centroid structure and what ensemble of RNA structures are authors referring to? Please explain what MFE is. not every reader knows this term.. Then Authors highlighted an extra hairpin loop in Figure 2 for mutants.. What could be the possible significance of this extra loop which is missing in natural structure? Are mutations happening here? Did the authors try to align the sequence for the natural and mutant? Please explain the outcomes from Figure 2 related to mountain plots and entropy plots... Authors talk about 3-bp CAA ins/del polymor-450 phism in 3'UTR of RTN4..Can they clarify if these INDELs cause change in mRNA structure in the middle of mRNA? Where is 5' and 3' sites? please mention in Figure 2.. what is ObservedBackground
Strength
FDR in Table 6? Nothing is mentioned in text...
It seems that computational studies are carried out just to add bulk to the manuscript. Please justify why mRNA and PPI network calculations are necessary and what are the inferences without just mentioning numbers and images obtained from tools
Extensive editing is required to make the results understandable.
Author Response
Although this is a case study for a specific population, which seems an interesting idea. However, the results are clearly not presented well. Extensive and careful editing is required to present the meaningful results providing supporting references. A few remarks ------------------
What is PPIN in Abstract?
This information is too much to give in Abstract" Our findings revealed that this variant was associated with decreased risk of BC 57 in heterozygous (OR=0.59, 95%CI= 0.39-0.89, p=0.012 D/I vs. D/D), dominant (OR=0.62, 95%CI=0.43- 58 0.91, p=0.015, D/I + I/I vs. D/D), over-dominant (OR=0.62, 95%CI=041-0.92, P=0.017, D/I vs. D/D-I/I) 59 and allelic (OR=0.72, 95%CI=0.52-0.99, p=0.035, I vs. D) models, while there were no significant dif- 60 ferences in homozygous (OR=0.74, 95%CI=0.38-1.46, p=0.388, I/I vs D/D), or recessive (OR=0.92, 61 95%CI=0.47-1.78, p=0.800, I/I vs I/I + D/D) models " Can you generalize it by highlighting a few points rather than detailed comments in the abstract..
Response: Thank you for the comment. The authors considered the constructive comment of the respected reviewer and revised the abstract. We have made it more general (lines 54-63).
In keywords: Keywords: RTN4, rs34917480, polymorphism, 5 What is the meaning of 5?
Response: Thank you for bringing this oversight to our attention. It was a typo mistake and we replaced it with “breast cancer”.
Can authors explain the details provided in Table 1? How is it relevant in this study?
Response: “Table 1” describes the potential interacting proteins with RTN4 in separate functional clusters based on existing evidence. For further clarification, we included an explanation for the network properties and statistics in subsection 2.3.2, lines 187-194, as follows:
‘… Table 1, in which nodes represent proteins or protein entities, edges-the interaction or connection between nodes, expected number of edges-expected edges based on a random model or null hypothesis, average node degree-the average number of connections each node has, Avg. local clustering coefficient-the tendency of neighboring nodes to form clusters, inflation parameter, the parameter used in the Markov Cluster Algorithm (MCL) for clustering analysis that influence the granularity and size of the resulting clusters, and enrichment p-value revealing the statistical significance of the observed clustering (see Supplementary file S1 for clusters detail).’
In Table 2: 49.30 (11.96) is SD of the age for control, and in text it is 11.97... Please be careful with all the numbers that authors report in text.
Response: Thank you for pointing out the error. We corrected the number in the Result section- line 208- and in Table 2. We also checked all numbers based on the respected reviewer’s comment.
What is DD DI II D I in Table 3, 4, 5? Not clear when we even read the text.
Response: Thank you for your feedback. In our initial submission, we used single letter abbreviation. In the revised manuscript, we changed all single-letter abbreviations to Del/Ins and included them in the description (and throughout the manuscript) to improve readability.
"The 500-bp region flanking the variant was used to evaluate the influence of the 227 rs34917480 (3-bp ins/del) on the RTN4 mRNA structure." What is the length of the variant? Please mention it.
Response: Thank you for the comment. The length of rs34917480 is 3 base-pair. Selecting the 500 bp region flanking the variant (extends 250 base pairs upstream and downstream of the variant site) is a common practice to ensure we included sufficient sequence context that could be influenced by the variant that might modulate gene expression or mRNA structure, such as potential regulatory elements, structural motifs, or functional regions (e.g., promoters, enhancers, or transcription factor binding sites) – included in lines 177-179, as follows:
‘A 500-bp region flanking the variant was analyzed to ensure including sufficient sequence context that could be influenced by the variant, potentially modulating RTN4 gene expression or mRNA structure.’
It is not clear why mRNA prediction is necessary here? Please explain carefully.
Response: Thank you for the suggestion. We aimed to assess any potential structural changes or alterations in the RTN4 mRNA molecule resulting from the existence of this 3-base pair polymorphism. This analysis helped to elucidate how (and to what extent) rs34917480 might influence the folding, stability, or other aspects of the mRNA structure, potentially impacting gene expression or functional properties of the RTN4 gene (by estimated changes in the free energy of the mRNA’s thermodynamic ensembles) – included in lines 179-182, as follows:
‘…the estimated influence of rs34917480 on the folding, stability, or other aspects of the mRNA structure, potentially impacting gene expression or functional properties of the RTN4 gene by estimated changes in the free energy of the mRNA’s thermodynamic ensembles.’
Which tool did the authors use to convert genes to mRNA structure first?
Response: As mentioned on page 4, line 173, we examined whether the RTN4 rs34917480 polymorphism can affect the secondary structure as well as the local folding of RTN4 mRNA using RNAfold, a web-based bioinformatics tool to predict RNA secondary structure (i.e., post-transcriptional) using deep learning- description included in the Methods, subsection 2.3.1. lines 174-177. Technical detail and more information about this tool are available in the corresponding paper of this tool on reference 37 of the manuscript (Sato et al., Nature 2021).
What is the meaning of this sentence? Our result demonstrated that the rs34917480 variant causes 231 changes in the MFE structure, the thermodynamic ensemble of RNA structures, and the 232 centroid structure. What is the centroid structure and what ensemble of RNA structures are authors referring to?
Response: This is the logic behind the secondary structure prediction using bioinformatics tools. Our result showed that rs34917480 could alter the thermodynamic properties of the RNA structure of RTN4/Nogo which can have consequences for the secondary structure, function and translational activies of the mRNA molecule. As illustrated in the Figure 2 and mentioned in lines 247-258, these changes include modifications in the minimum free energy (MFE) structure -the energetically most stable conformation of the RNA molecule, the thermodynamic ensemble of RNA structures – the range of possible conformations the RNA can adopt under different conditions, and the centroid structure -the central or representative structure within the ensemble. The altered thermodynamic ensembles can make certain secondary structures more or less favorable, potentially leading to shifts in the mRNA's folding patterns, stability, and overall structure. Consequently, these structural changes have the potential to influence the mRNA's secondary structure, function and translational activities, which can impact processes such as gene regulation, protein synthesis, and cellular functions. According to the RNAfold, the centroid structure of an RNA sequence is the secondary structure with minimal base pair distance to all other secondary structures in the Boltzmann ensemble. In comparison with the MFE structure, the centroid of the ensemble makes 30.0% fewer prediction errors as measured by the positive predictive value (PPV) with marginally improved sensitivity. More technical detail can be found in a paper by Ding, Chan and Lawrence 2005 (PMID: 16043502).
As the scope of our article did not encompass that particular explanation, we made the decision to exclude it. However, if you still believe it would enhance the article's clarity, we are open to including it based on your suggestion.
Please explain what MFE is. not every reader knows this term.
Response: We explained MFE in our response to the previous question. Briefly, the MFE structure of an RNA sequence is the secondary structure that contributes a minimum of free energy. This structure is predicted using a loop-based energy model and the dynamic programming algorithm introduced by Zuker and Stiegler, 1981 (PMID: 6163133).
Also, we added a definition of MFE to section 2.3.1, lines 174-177 as follows:
‘…structure by free energy minimization (i.e., minimal free energy [MFE] structure). The MFE is defined as “the secondary structure that contributes a minimum of free energy.’
Then Authors highlighted an extra hairpin loop in Figure 2 for mutants. What could be the possible significance of this extra loop which is missing in natural structure? Are mutations happening here? Did the authors try to align the sequence for the natural and mutant? Please explain the outcomes from Figure 2 related to mountain plots and entropy plots...
Response: Thank you for your comment. In general, there is no “natural” vs. “mutant” in polymorphic studies. Although we can call certain variants as “wild” type, it is merely based on the frequency of such variants in that specific population. The plot visualizes the changes caused by the variant in terms of the minimal free energy (MFE), thermodynamic ensemble (PF), and centroid structures. The slopes, plateaus, and peaks in the plot correspond to different regions of the RNA structure- helices (slopes), loops (plateaus), and hairpins (peaks). The free energy of the thermodynamic ensemble for insertion and deletion was -252.68 and -251.85 kcal/mol, respectively, which resulted in higher entropy in these mentioned regions. This higher entropy in turn reflects the higher variability of RNA conformations within the thermodynamic ensemble and is associated with increased instability. Notably, the presence of an extra hairpin loop in the figure further provides a structural alteration caused by the variant – explanation included in lines 248-252, as followed:
‘The free energy of the thermodynamic ensemble for insertion and deletion was -252.68 and -251.85 kcal/mol, respectively, which resulted in higher entropy in specific regions shown in the Figure 2. This higher entropy in turn reflects the higher variability of RNA conformations within the thermodynamic ensemble and is associated with increased instability’.
Authors talk about 3-bp CAA ins/del polymorphism in 3'UTR of RTN4. Can they clarify if these INDELs cause change in mRNA structure in the middle of mRNA? Where is 5' and 3' sites?
Response: The 3' untranslated region (3’UTR) is a segment of the messenger RNA (mRNA) that follows the coding sequence and terminates the transcript. The 3'UTR plays a crucial role in post-transcriptional regulation, including mRNA stability, localization, and interaction with regulatory molecules. This “CAA” 3 base pair insertion/ deletion is located in 3’UTR of RTN4.
please mention in Figure 2. what is Observed.
Please explain
Background
Strength
FDR in Table 6? Nothing is mentioned in text...
Response: Observed refers to the number of proteins that RTN4 PPIN network is associated with, background to the total number of proteins in the PPIN network that RTN4 can potentially be associated with, strength to the significance of the association between the proteins in the PPIN network and the specific functional category and false discovery rate (FDR) is a statistical measure that quantifies the proportion of false positive results among the total significant findings. FDR is a commonly used method for correcting type I error in multiple comparisons, just like other statistical correction, such as Bonferroni correction. The explanation is included in subsection 3.2.2. lines 275-279 and in Table 6’s description.
It seems that computational studies are carried out just to add bulk to the manuscript. Please justify why mRNA and PPI network calculations are necessary and what are the inferences without just mentioning numbers and images obtained from tools
Response: We appreciate your concern and feedback. The oncogenic roles of RTN4/Nogo have been investigated in breast cancer; several mechanisms have been also suggested for the association of RTN4 with BC. Using mRNA secondary structure prediction, our results can offer a deeper understanding of complex phenomena by providing mechanistic insights into how rs34917480 may influence RTN4 expression and molecular processes related to BC development and progression. Throughout the literature, only one study (reference 88, G. Novak, 2006) has suggested that this variant can dysregulate RTN4/Nogo isoforms expression, leading to several diseases, particularly cancers - our study rather provides a more comprehensive, robust analysis and broader perspective on RTN4 association with cancer development, as discussed in the Discussion, lines 435-436 and 460-464.
Furthermore, by analyzing the PPIN of RTN4, we aimed to explore and interpret complex data, analyze large-scale datasets, or simulate biological processes that are otherwise challenging to study solely through experimental means. In addition to the suggested molecular bases of RTN4/Nogo in BC carcinogenesis by previous studies (such as interaction with apoptosis as well as PI3/AKT and MAPK/ERK pathways), our predicted PPIN indicated several other genes, proteins, cellular processes and signaling pathways associated with BC that are less mentioned in RTN4 studies, as discussed in the Discussion, paragraphs 4-7 and lines 345-427.
By solely relying on numbers and images from mRNA secondary structure prediction and PPIN analysis, we could run the risk of lacking the context and interpretation, the limited information provided, potential bias towards specific tools, incomplete interpretation of complex biological systems and the need for further experimental validation. To address these detriments, it is crucial to provide context, detailed methodology descriptions, statistical analyses, validation experiments, and critical interpretation alongside the presentation of numbers and images
We acknowledge the importance of balancing computational and experimental approaches and have taken steps to ensure that the computational analyses presented in the manuscript are relevant, well-justified, and contribute meaningfully to the overall research findings. By employing both computational and experimental methods, we have delved into the sophisticated details of these mechanisms, shedding light on the underlying factors that contribute to the observed decrease in BC risk associated with rs34917480. Our results suggest that rs34917480 might be associated with higher expression of RTN4/Nogo isoforms (based on mRNA secondary structure prediction), enhancing their pro-apoptotic activity that suppresses tumor development which might reduce or overcome the oncogenic role of RTN4/Nogo isoforms, which is highly likely induced by PI3K/AKT and MAPK/ERK activation (based on PPIN analysis)-explained in the Discussion, lines 454-464. We hope that our results provide a more comprehensive understanding of the molecular mechanisms, regulatory networks and potential biomarkers associated with RTN4’s rs34917480 and decreased risk of BC. We believe that these findings not only contribute to the current body of knowledge but also lay a solid foundation for future investigations in the field of breast cancer research, with the ultimate goal of advancing diagnosis, treatment, and personalized medicine approaches- further clarified in the introduction, lines 115-117. Thank you for raising this concern, and we are open to further discussing and addressing any specific questions or clarifications you may have.
Reviewer 3 Report
life- 2418108
In this case control study titled “The Reticulon-4 3-bp Deletion/Insertion Polymorphism is Associated with mRNA Structural Changes and Risk of Breast Cancer: A Population-based Case-control Study with Bioinformatics Analysis” by Pourzand et al., the authors aimed to investigate the association between a genetic variant of Reticulon-4 (rs34917480) and the susceptibility of breast cancer in an Iranian subpopulation. In this study, majority of the work were focused on 3 base pair indels at the 3’ UTR of the reticulon 4 from healthy and breast cancer patient’s blood. Using Web based bioinformatic tools to identify the structural changes of reticulon-4 genetically variant mRNA adds up a little but deviates the whole purpose of the study. Experimental design to provide enough evidence for the argument is not outstating, however, this study has statistically excellent number of patients and healthy controls but stands to gain from the following concerns.
Major:
Authors claims that their findings may be used in future research to develop diagnostics and treatment options.
Authors must provide enough evidence to support their insertion/deletion polymorphism data when studying a genetic variant that has not been reported in breast cancer before. For example, random (at least from 5 patients and healthy subjects) sequencing of PCR products to confirm the insertion or deletion is strongly recommended. False positive results in agarose gel electrophoresis of PCR products have been evidenced in the past.
I request authors to clearly present the results. For example, gel electrophoresis. Author claims that the presence of an allele-specific band of 308 bp with a control band of 212 bp size was considered positive evidence for each allele. Lane 2: Deletion/Deletion but bands 308 bp bands are not present in both the lanes along with internal controls. It is really difficult to understand the results. Please limit the use of single letter abbreviations.
Minor:
Please provide details on the methodology of specific primer design
Was whole blood stored at -20 0C? As per many others and our own study, we always store whole blood at 40C for few hours before processing and the plasma either at -20 or -80 until further use.
Why have authors provided details of only forward primers for both insertion and deletion allele?
Why have authors used pri-miR-34b/c as internal reference control instead of standard reference controls?
Good
Author Response
In this case control study titled “The Reticulon-4 3-bp Deletion/Insertion Polymorphism is Associated with mRNA Structural Changes and Risk of Breast Cancer: A Population-based Case-control Study with Bioinformatics Analysis” by Pourzand et al., the authors aimed to investigate the association between a genetic variant of Reticulon-4 (rs34917480) and the susceptibility of breast cancer in an Iranian subpopulation. In this study, majority of the work were focused on 3 base pair indels at the 3’ UTR of the reticulon 4 from healthy and breast cancer patient’s blood. Using Web based bioinformatic tools to identify the structural changes of reticulon-4 genetically variant mRNA adds up a little but deviates the whole purpose of the study. Experimental design to provide enough evidence for the argument is not outstating, however, this study has statistically excellent number of patients and healthy controls but stands to gain from the following concerns.
Major:
Authors claims that their findings may be used in future research to develop diagnostics and treatment options.
Response: We appreciate your comments and feedback. In general, polymorphic studies are a great opportunity to study individual differences, and to tailor diagnosis and treatment approaches based on these individual differences for personalized medicine. Certain polymorphic studies could help to improve diagnostic biomarkers, or access response to treatment in a specific population, yet, further studies with larger dataset and more complex methods are needed to confirm these results. Some previous studies have suggested a link between Nogo-B (one of three major RTN4’s isoforms) expression, and breast cancer survival as well as response to tamoxifen and paclitaxel, signifying its prognostic and therapeutic potential. In line with this, our results suggest that rs34917480 might be associated with higher expression of RTN4/Nogo isoforms, enhancing their pro-apoptotic activity that suppresses tumor development that might reduce or overcome the oncogenic role of RTN4/Nogo isoforms, which is highly likely induced by PI3K/AKT and MAPK/ERK activation. These mechanisms, as mentioned in the Results, subsection 3.2.2. as well as Table 6 and further explored in the Discussion – paragraphs 4-7, provide a foundation for further research to develop diagnostics and treatment options for breast cancer sufferers by aiding in understanding the functional consequences, identifying biomarkers, discovering thorough therapeutic targets and promoting personalized medicine approaches. Our study strives to provide a detailed understanding of the molecular mechanisms, regulatory networks, and potential biomarkers linked to the RTN4’s rs34917480 that lowers breast cancer risk, since our goal, as always, is to provide a thorough and well-rounded scientific study that advances knowledge in the field. We believe that these findings not only contribute to the current body of knowledge but also lay a solid foundation for future investigations in the field of breast cancer research, with the ultimate goal of advancing diagnosis, treatment, and personalized medicine approaches- further clarified in the introduction, lines 115-117. Thank you for raising this concern, and we are open to further discussing and addressing any specific questions or clarifications you may have.
Authors must provide enough evidence to support their insertion/deletion polymorphism data when studying a genetic variant that has not been reported in breast cancer before. For example, random (at least from 5 patients and healthy subjects) sequencing of PCR products to confirm the insertion or deletion is strongly recommended. False positive results in agarose gel electrophoresis of PCR products have been evidenced in the past.
Response: Thank you for the comment. Aligned with your concerns, we tried to address and minimize the occurrence of any false positive results by optimizing PCR conditions and maintaining quality control by reproducibility. We randomly repeated genotyping for ~10% of samples for quality control, and all results were 100% reproduced. Also, we did sequencing for each genotype, as a standard routine that we used in our previous publications from our lab, which the result was concordant with AS-PCR results. We addressed such issues in Method and Result section (as follows) and added the sequencing result as the Supplementary figure (Supplementary figure S1) to the manuscript.
Method section (subsection 2.2, lines 157-161):
‘To ensure quality assessment, we randomly re-genotyped 10% of the samples to show the reproducibility of the genotyping results. Additionally, to confirm the AS-PCR genotyping results, we randomly selected DNA samples from AS-PCR amplification for DNA sequencing (n = 4 for each genotype; see the result section and supplementary figure S1).’
Result section (subsection 3.1., lines 225-229):
‘We randomly re-genotyped 10% of the samples as part of our quality control process for genotyping and the results demonstrated 100% reproducibility and absence of errors in the findings. Also, selected amplified DNA products of each genotype (n=4 for each genotype) were examined by DNA sequencing, which further confirmed our AS-PCR results (see Supplementary Figure S1 for sequencing results).’
I request authors to clearly present the results. For example, gel electrophoresis. Author claims that the presence of an allele-specific band of 308 bp with a control band of 212 bp size was considered positive evidence for each allele. Lane 2: Deletion/Deletion but bands 308 bp bands are not present in both the lanes along with internal controls. It is really difficult to understand the results. Please limit the use of single letter abbreviations.
Response: We acknowledge your concerns regarding the clarification of the result and method. To further clarify the results, the presence of the 212 bp bands in all lanes indicates that the PCR amplification is done successfully. The absence of the 308 bp bands in Lane 2 and Lane 3 indicates the sample is homozygous for that particular genotype (Lane 2 for deletion and 3 for insertion). We also changed all single-letter abbreviations to Del/Ins throughout the manuscript to improve clarity. Thus, we added extra explanation regarding the method as follows:
Method section (subsection 2.2., lines 142-150):
‘For this reaction, two allele-specific forward primers were designed -one for deletion allele (F1) and the other for insertion allele (F2)- along with one consensus generic reverse primer. Upon completion of PCR, two equally long bands were produced, with an amplicon size of 308 bp for both insertion and deletion -the presence of each band represents the presence of the corresponding allele [35, 36]. The forward and reverse primers of pri-miR-34b/c (5′-CCTCTGGGAACCTTCTTTGACCTGT-3′ and 5′-CCTGGGCCTTCTAGTCAAATAGTGA-3′, respectively) were used as the internal control and added to each tube to ensure successful amplification, which was indicated by the presence of 212 bp bands for the internal control in all samples.’
And
Figure 1 caption:
‘Figure 1. Gel electrophoresis. Photograph of electrophoresis pattern of the allele-specific PCR to detect RTN4 rs34917480. Each sample has two tubes, containing either insertion or deletion primer (along with reverse and internal control primers in both tubes). Lane 1: Insertion/Deletion, which is the result of the existence of both alleles for that sample (heterozygous); Lanes 2: Deletion/Deletion, which depicts the presence of only deletion allele for the sample (homozygous); Lane 3: Insertion/Insertion, which depicts the presence of only insertion allele for the sample (homozygous); product size for Insertion or Deletion and internal control is 308 bp and 212 bp, respectively. M: DNA marker.’
Please provide details on the methodology of specific primer design:
Response: We used conventional methods for primer design (done by the late Prof. Mohammad Hashemi), using BLAST tool and CLC Genomics software, and followed conventional guidelines for primer design (for instance, see guidance here https://medicine.yale.edu/keck/dna/protocols/guidelines/ and here https://ocw.mit.edu/courses/7-15-experimental-molecular-genetics-spring-2015/b6d9befecddfc6f51fc98157769dceaa_MIT7_15S15_Primer_design.pdf).
Such methodology has been employed for designing all primers for our previous publications. Importantly, since we have no restriction enzyme site for this experiment, primers used for this study is designed for the detection of specific allele on amplicons, making it a proficient way for detecting desired sequences. We also provided the primers’ sequence and also, further sequenced selected PCR products for each genotype to ensure our designed primers stably detected each genotype by AS-PCR method.
Was whole blood stored at -20 0C? As per many others and our own study, we always store whole blood at 40C for few hours before processing and the plasma either at -20 or -80 until further use.
Response: Thanks for the comment. As the reviewer correctly mentioned, storing blood samples at -20 is used for long-term storage, in case the samples are not going to be used for DNA extraction immediately, but samples were transferred to 4°C prior to extraction. We edited the corresponding explanation in the method section (Subsection 2.2., line 136):
‘The whole blood samples of patients and controls were collected in tubes containing EDTA and stored at -20 °C until further use, and moved to 4 °C prior to DNA extraction.’
Why have authors provided details of only forward primers for both insertion and deletion allele?
Response: Thank you for pointing out the oversight. The generic reverse primer is now included in the manuscript: 5`ACCGGTAAAGCAGGAATGACAA-3`- included in line 141, which was mistakenly removed from the submitted manuscript.
Why have authors used pri-miR-34b/c as internal reference control instead of standard reference controls?
Response: While standard reference controls, such as housekeeping genes, are commonly used, the selection of alternative internal reference controls, like pri-miR-34b/c, can offer certain advantages in terms of stability, biological relevance, consistency across samples, miRNA regulation, and specificity. We previously used pri-miR-34 b/c and thus were certain regarding its stability and
Round 2
Reviewer 2 Report
Although the authors tried to reply to all the comments, I will suggest to include all the justifications in the edited version of manuscript rather than including it in 'reply to reviewers".
English is fine but careful proofreading is recommended.
Author Response
Thank you for your feedback on our manuscript. We appreciate your suggestions and we agree that consolidating all the justifications within the manuscript itself would enhance the clarity and accessibility of our work. Thus, we have incorporated the necessary justifications and explanations within the relevant sections of the paper, ensuring that readers can easily understand the reasoning behind our choices and conclusions. In response to your comment, we have revised the manuscript accordingly, as highlighted in the manuscript text and follows:
- The suggestion for simplifying and generalizing the abstract is addressed in lines 54-59 where we defined any abbreviations used in the abstract to ensure that readers can easily understand the context and revised the wording to generalize the results, making them more broadly applicable and relevant to a wider audience.
- Thanks to your attention, the oversight in the keywords is corrected, and a new keyword (i.e., breast cancer) is added in line 65.
- We have considered your suggestion regarding the contents of Table 1 and addressed it in subsection 2.3.2, specifically in lines 187-194. In this revised section, we have provided additional explanations and made modifications to clarify what the contents of Table 1 represent and what they indicate. We should reiterate that this Table shows the statistics used to build the PPIN, which needs to be reported as part of scientific methodology in terms of clarification and reproducibility of the results.
- The error of mean and SD in Table 2 and line 209 is corrected, thanks to you bringing the oversight to our attention.
- We changed all ID, II, and DD abbreviations of genotypes to other common yet easy-to-read forms, as they are Del/Ins, Ins/Ins, and Del/Del in Tables 3-5 and throughout the entire manuscript.
- Building upon your feedback, we have provided additional clarification regarding the mention of a 500 bp flank in lines 177-179 and 247-248. In those sections, we have explained the significance of the 500 bp flank in the context of our study. referring to the surrounding genomic region on either side of the three-base pair variant.
- Because all polymorphisms are of importance when they affect cell biology (particularly in protein-coding genes where 3'UTR polymorphisms can impact protein production, as we included in lines 101-103), we have further clarified the logic and rationale behind conducting the mRNA prediction analysis in lines 179-182 and expanded the discussion in lines 463-467 in response to your comment. We also emphasized that investigating the potential impact of the 3'UTR polymorphism on mRNA stability, secondary structure, or binding sites can provide valuable insights into its functional significance in lines 258-261.
- With your suggestion, we have revisited the method section and made additional improvements to clarify the mRNA structure prediction method in the method section, lines 174-177, adding on top of the previously described method for mRNA structure, including the software. We have also provided further explanation and additional details to ensure that readers can easily understand the approach we employed, including the specific software utilized. Additionally, we have added an extra explanation regarding the centroid structure in lines 256-258 of the manuscript. The explanation of the minimum free energy (MFE) is now provided in lines 175-177. Lastly, to disambiguate the explanation regarding Figure 2, we have revised the relevant section in lines 249-254. Thanks to your feedback, we hope that these modifications aim to provide a more precise and comprehensive understanding of the content and purpose of the figure.
- Thanks to your feedback, we have included the explanation for the terminology used for Table 6 as a table footnote and in lines 278-282.
- Upon careful review, we included a comprehensive explanation that outlines the rationale, methodology and implications of the bioinformatic analysis conducted in different sections of our study in the following addresses: lines 115-117, lines 435-436 and 460-464, paragraphs 4-7 (lines 345-427), as well as lines 454-464.
We appreciate the reviewer’s intention to improve the clarity of our manuscript. We have made every effort to carefully address each point and provide comprehensive explanations and clarifications where necessary and would appreciate the reviewer’s observation of the detailed explanations throughout the manuscript. With thorough proofreading, we, therefore, aimed to address all concerns the reviewer raised; however, if still any specific issue remains to be further explained, we are willing to address it.
Reviewer 3 Report
I appreciate the effort from authors in addressing all the concerns raised.
Author Response
We appreciate the reviewer’s comments, which improved the quality of our manuscript.